# Two-Step Calcination-Method-Derived Al-Substituted W-Type SrYb Hexaferrites: Their Microstructural, Spectral, and Magnetic Properties

**Yujie Yang**

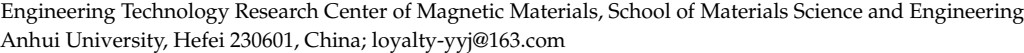

Engineering Technology Research Center of Magnetic Materials, School of Materials Science and Engineering, Anhui University, Hefei 230601, China; loyalty-yyj@163.com

**Abstract:** W-type hexaferrites were discovered in the 1950s and are of interest for their potential applications. In this context, many researchers have conducted studies on the partial substitution of Fe sites in order to modify their electric and magnetic properties. In this study, W-type SrYb hexaferrites using $Al^{3+}$ as substitutes for $Fe^{3+}$ sites with the nominal composition $Sr_{0.85}Yb_{0.15}Zn_{1.5}Co_{0.5}Al_xFe_{16-x}O_{27}$ ($0.00 \leq x \leq 1.25$) were successfully synthesized via the two-step calcination method. The microstructures, spectral bands of characteristic functional groups, morphologies, and magnetic parameters of the prepared samples were characterized using XRD, FTIR, SEM, EDX, and VSM. The XRD results showed that, compared with the standard patterns for the W-type hexaferrite, the W-type SrYb hexaferrites with the Al content (x) of $0.00 \leq x \leq 1.25$ were a single-W-type hexaferrite phase. SEM images showed the flakes and hexagonal grains of W-type hexaferrites with various Al content (x). The saturation magnetization ($M_s$) and magneton number ($n_B$) decreased with Al content (x) from 0.00 to 1.25. The remanent magnetization ($M_r$) and coercivity ($H_c$) decreased with Al content (x) from 0.00 to 0.25. Additionally, when the Al content (x) $\geq 0.25$, $M_r$ and $H_c$ increased with the increase in the Al content (x). The magnetic anisotropy field ($H_a$) and first anisotropy constant ($K_1$) increased with the Al content (x) increasing from 0.00 to 1.25. Al-substituted W-type SrYb hexaferrites with soft magnetic behavior, high $M_s$, and lower $H_c$ may be used as microwave-absorbing materials.

**Keywords:** two-step calcination method; W-type hexaferrites; magnetic properties; X-ray diffraction

## 1. Introduction

The hexaferrites of W-type compounds with the chemical formula $AMe_2Fe_{16}O_{27}$ (in this formula, A stands for the alkaline earth metal ion, and Me stands for the bivalent transition element ion) were discovered in the 1950s and are of interest for their potential applications [1]. Due to their strong anisotropy and moderate magnetization, W-type hexagonal ferrites have become important magnetic materials. In recent years, the scientific community is keen to develop W-type hexagonal crystals, as manifested by their application in high-density magnetic recording, high-performance permanent magnets, microwave materials, microwave-absorbing materials, etc. [2]. Various synthesis methods have been used to prepare W-type hexaferrites. Some methods have been reported, such as the glass crystallization method [3], the sol–gel method [4], the hydrothermal method [5], a combined method using molten salt and sol–gel [6], the traditional ceramic method [7], the sol–gel self-combustion method [8], chemical coprecipitation [9], and chemical solution deposition [10]. By comparing these methods, it is found that the advantages of the traditional ceramic method are as follows: simplicity, controllable grain size, high productivity, etc.

Recently, many researchers have conducted studies on the partial substitution of Fe sites in order to modify their electric and magnetic properties [11–14]. Albanese et al. reported the cation distribution and magnetic properties of $BaZn_2Fe_{16-x}In_xO_{27}$ and $BaZn_2Fe_{16-x}Sc_xO_{27}$ hexagonal ferrites prepared by using the standard ceramic technique, and they found that the Sc-containing samples ($x \leqq 2$) had no evidence of iron substitution at 2d position; in

the indium series, where the measurement range was extended to x = 4.5, the In ions did not enter the hexahedral position of x = 3.5, and in the In-substituted compounds, at the substitution (x) $\geq$ 2, the slope of $T_C$ and x became steeper, and a strong dependence of magnetization ($\sigma$) on the external magnetic field was observed above and below $T_C$ [11]. The effects of the Sc ion substitution for Fe ions on the resulting microstructure and magnetic properties were investigated by Qi et al. Their results showed that, with the increase in Sc substitution, the squareness ratio ($M_r/M_s$) of the samples ranged from 0.78 to 0.82, $H_a$ decreased, and $M_s$ increased from 53.61 to 54.04 emu/g with the increase in x from 0 to 0.1, but when x > 0.1, $M_s$ was reduced to 46.50; in addition, the sample also had other excellent magnetic properties, namely an appropriate anisotropy field (about 9 kOe) and high coercive fields (about 1300 Oe) [12]. Ahmadet et al. synthesized the lanthanum-doped W-type hexaferrites $BaZn_2La_xFe_{16-x}O_{27}$ (x = 0, 0.2, 0.4, 0.6, 0.8, 1.0) via coprecipitation and sintered the samples at 1320 °C to study the structural, magnetic, and electrical properties of $BaZn_2La_xFe_{16-x}O_{27}$ powders. They found that, as La substitution increased, the grain size decreased due to the fact that La acts as an inhibitor. The decrease in saturation magnetization and remanence is attributed to the spin tilt at the B-site, and the coercivity increase follows 1/r, where r is the radius of the particle. Furthermore, the DC resistivity increased from $0.59 \times 10^7$ $\Omega$ cm to $8.42 \times 10^7$ $\Omega$ cm with the increase in La substitution because of the unavailability of $Fe^{3+}$ ions [13]. The impact of Dy addition on the structural, morphological, electrical, and magnetic properties of nanocrystalline W-type hexaferrites prepared using the citrate sol–gel method was reported, and the results exhibited that, with the increase in Dy content, the values of coercivity ranged between 530 and 560 Oe; the magnetization declined as the Dy content increased; a minimum reflection loss of $-40$ dB was observed at 16.2 GHz for the samples with x = 0.6 and 1.7 mm thickness, and as Dy content increased, the minimum reflection point moved toward high frequency [14]. Aen et al. prepared a series of nanosized single-phase W-type $SrGa_xZn_2Fe_{16-x}O_{27}$ (x = 0.0, 0.1, 0.2, 0.3, 0.4) hexaferrites via the sol–gel technique and investigated the role of Ga substitution on the magnetic and electromagnetic properties of nanosized W-type hexagonal ferrites. They found that as the Ga content (x) increased, the saturation magnetization ($M_s$) decreased, and the coercivity ($H_c$) was enhanced. All the samples had $H_c$ values in the range of hundreds of oersteds. In the frequency range of 0.5–13 GHz (relative to $-20$ dB), the microwave absorption performance was improved, and the bandwidth could reach 0.899 GHz [15]. The influence of Al substitution on the magnetic, electrical, and high-frequency dielectric constants of $BaCo_2Al_xFe_{16-x}O_{27}$ (x = 0–1) hexaferrites was investigated by Ahmad et al., and the results showed that due to the unavailability of $Fe^{3+}$ ions, the DC resistivity enhanced as the Al content increased; the coercivity ($H_c$) values of all the samples were in the range of several hundred oersteds. Thus, this is one of the necessary conditions for their use in electromagnetic materials, as well as for safety, switching, and sensing applications; in the frequency range of 0.025–1.5 GHz, the high-frequency permittivity and dielectric loss decreased with the increase in the Al content [16]. Cr-doped W-type hexagonal ferrites with the composition $BaNi_2Cr_xFe_{16-x}O_{27}$ ($0.0 \leq x \leq 0.4$) were synthesized using the sol–gel auto-combustion technique, and the structural, magnetic, and dielectric properties of the Cr-substituted W-type hexaferrites were investigated by Rehman et al. Based on their results, the SEM images revealed a platelet-like and hexagonal shape; the hysteresis loops of all the samples were thin and had low coercivity (about a few hundred oersteds) and exhibited soft magnetic properties. This is a prerequisite for high-frequency applications, and in the frequency range of 1–6 ghz, the high-frequency dielectric constants follow Koop's phenomenological theory, which is consistent with the Maxwell-Wegner model [17]. Iqbal et al. prepared the nanocrystalline W-type hexaferrites of the nominal stoichiometry of $BaNi_2Cr_xFe_{16-x}O_{27}$ (x $\leq$ 1) using the chemical coprecipitation method and investigated their structural, electrical, and magnetic properties. They found that, when Cr was incorporated into $BaZn_2Fe_{16}O_{27}$, the room temperature resistivity ($\rho_{RT}$) for $BaZn_2Fe_{16}O_{27}$ increased to the order of $10^9$ $\Omega$ cm, which is higher than that of W-type hexaferrites ($10^3$–$10^7$ $\Omega$ cm) previously reported; the dielectric

constant ($\varepsilon'$) and dielectric loss ($\tan \delta$) decreased with the increase in frequency, and both of them increased with the increase in Cr doping; the improvement in the above parameters results in them having a broad application prospect in high-frequency radar absorption and electromagnetic interference attenuation [18].

In the present work, the modified ceramic method, i.e., the two-step calcination method, was used to prepare the samples. Additionally, Al-substituted W-type SrYb hexaferrites were fabricated with the nominal composition $Sr_{0.85}Yb_{0.15}Zn_{1.5}Co_{0.5}Al_xFe_{16-x}O_{27}$ ($0.00 \leq x \leq 1.25$) via the two-step calcination method. Fe is a strong magnet, and its substitution with Al, which is a weak magnet, modifies the properties of the synthesized W-type SrYb hexaferrites. Thus, the effects of the Al substitution on the microstructure, spectral, and magnetic properties of W-type SrYb hexaferrites were investigated. The novelty of this paper is that the Al-substituted W-type SrYb hexaferrites were synthesized via the two-step calcination method for the first time, and tunable $M_s$ and $H_c$ with Al substitution were obtained.

## 2. Materials and Methods

### 2.1. Sample Preparation

In this work, $SrCO_3$ (99.5 wt.%) powder, $Yb_2O_3$ (99.9 wt.%) powder, $ZnO$ (99 wt.%) powder, $CoO$ (99 wt.%) powder, $Al_2O_3$ (99 wt.%) powder, $Fe_2O_3$ (99.3 wt.%) powder were used as raw materials. All the reagents used in this paper were bought from Aladdin Industrial Corporation. All the starting materials were used as received, i.e., no further chemical decontamination was carried out. W-type SrYb hexaferrites using $Al^{3+}$ as substitutes for $Fe^{3+}$ sites with the nominal composition $Sr_{0.85}Yb_{0.15}Zn_{1.5}Co_{0.5}Al_xFe_{16-x}O_{27}$ ($0.00 \leq x \leq 1.25$) were successfully synthesized via the two-step calcination method. In this article, in the nominal composition $Sr_{0.85}Yb_{0.15}Zn_{1.5}Co_{0.5}Al_xFe_{16-x}O_{27}$ ($0.00 \leq x \leq 1.25$), the subscript of each metal element and oxygen is the ratio of the number of moles. The starting materials were weighed in a stoichiometric composition and were wet-mixed in water for 3 h in a planetary ball mill, with a rotation velocity of 300 rpm and a ball-to-powder of about 6:1. Further, the ground powder was dried and compressed into round pellets. In the first step, the as-prepared pellets were calcined in air for 2.0 h in a muffle furnace at 1300 °C. The heating rate was maintained at 4 °C/min, and cooling was achieved by keeping the furnace closed and cooling the furnace to room temperature. The calcined pellets were pulverized to powder using a vibration mill. In the second step, the pulverized powder was compressed into round pellets and re-calcined in air for 2.0 h in a muffle furnace at 1300 °C. In order to obtain the fine powders, the calcined pellets were crushed using a vibration mill. In order to eliminate the crystal strain and lattice defects, the crushed powder was annealed in air at 800 °C for 2.5 h in a muffle furnace.

### 2.2. Techniques of Characterization

The X-ray diffraction (XRD) study of all the W-type SrYb hexaferrite powders was carried out by using a Rigaku X-ray diffractometer equipped with Cu $K_\alpha$ ($\lambda = 1.5406$ Å) radiation. The infrared spectra were recorded using a Fourier transform infrared (FTIR) spectrometer (Nicolet 6700, Thermo Scientific, Waltham, MA, USA) for all the samples with KBr pellets in the wavenumber range of 4000 to 400 $cm^{-1}$. The morphology for all the samples was detected using a field-emission scanning electron microscope (FE-SEM, Hitachi S-4800, Tokyo, Japan). The elemental compositions of the samples were detected with an energy-dispersive X-ray spectrometer (EDX). The magnetic hysteresis loops for all the samples were measured by using a vibrating-sample magnetometer (VSM) (VSM 3100, Beijing Oriental Chenjing Technology Co., Ltd., Beijing, China) at room temperature under an external magnetic field of 24,000 Oe.

## 3. Results and Discussion

### 3.1. XRD Analysis

Figure 1 shows the XRD patterns of the W-type SrYb hexaferrite $Sr_{0.85}Yb_{0.15}Zn_{1.5}Co_{0.5}Al_xFe_{16-x}O_{27}$ ($0.00 \leq x \leq 1.25$). As shown in Figure 1, the XRD patterns confirmed that all the samples had a single-phased W-type hexaferrite structure, and the peaks of all the samples matched well with the standard JCPDS card (no. 75-0406); no impurity or undesirable phase was observed for any sample. This means that Al substitution does not change the magnetoplumbite structure of W-type SrYb hexaferrites.

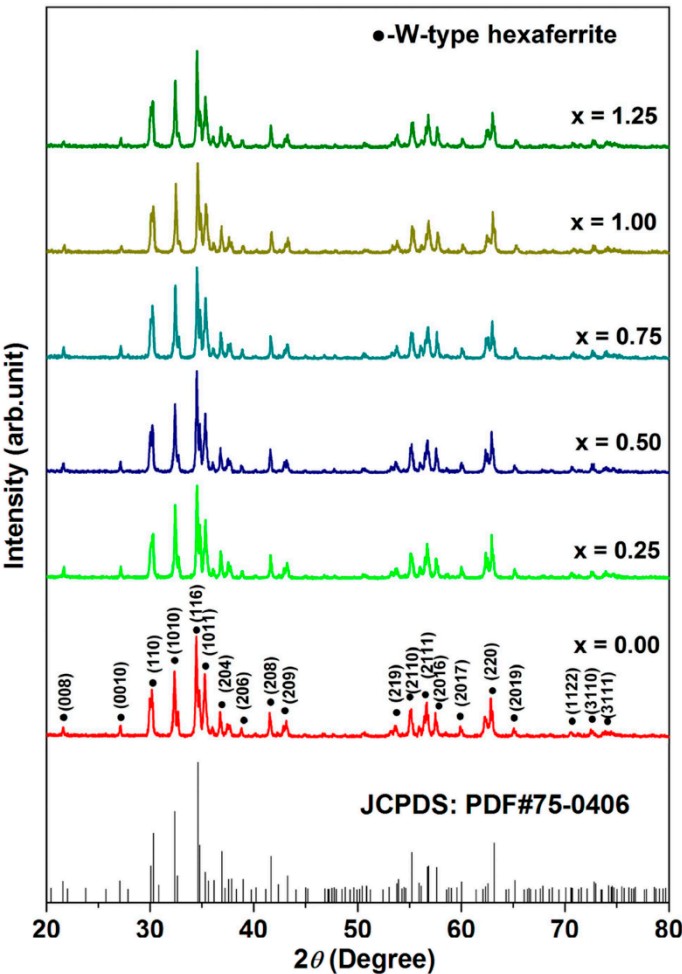

**Figure 1.** The XRD patterns of the W-type SrYb hexaferrite $Sr_{0.85}Yb_{0.15}Zn_{1.5}Co_{0.5}Al_xFe_{16-x}O_{27}$ ($0.00 \leq x \leq 1.25$).

The lattice constants $c$ and $a$ for the W-type hexagonal ferrites were calculated from the XRD data by using the following formula [19]:

$$\frac{1}{d_{hkl}^2} = \frac{4}{3} \times \frac{h^2 + hk + k^2}{a^2} + \frac{l^2}{c^2} \tag{1}$$

where $d_{hkl}$ is the inner planner spacing of the XRD pattern, and $h$, $k$, and $l$ are Miller exponents. The lattice constants ($c$ and $a$) for the W-type SrYb hexaferrite $Sr_{0.85}Yb_{0.15}Zn_{1.5}Co_{0.5}Al_xFe_{16-x}O_{27}$ ($0.00 \leq x \leq 1.25$) are depicted in Figure 2. It can be clearly seen from the graph that the values of $c$ and $a$ of W-type SrYb hexagonal ferrites with Al substitution ($0.25 \leq$ Al content $(x) \leq 1.25$) are smaller than that of W-type SrYb hexagonal ferrites without Al substitution. The decrease in lattice parameters $c$ and $a$ for the Al-substituted W-type SrYb hexaferrites ($0.25 \leq x \leq 1.25$) is because the ionic radius of $Al^{3+}$ (0.535 Å) is smaller than that of $Fe^{3+}$

(0.645 Å) [20–22]. However, as the Al content (x) increased from 0.25 to 1.25, the lattice parameters *c* and *a* of the Al-substituted W-type SrYb hexagonal ferrites fluctuated within a certain range.

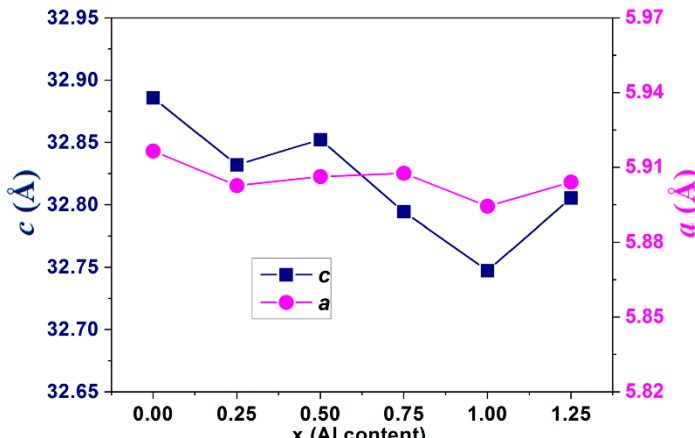

**Figure 2.** Lattice constants (*c* and *a*) for the W-type SrYb hexaferrite $Sr_{0.85}Yb_{0.15}Zn_{1.5}Co_{0.5}Al_xFe_{16-x}O_{27}$ ($0.00 \leq x \leq 1.25$).

### 3.2. FT-IR

The FT-IR spectra of the W-type SrYb hexaferrite $Sr_{0.85}Yb_{0.15}Zn_{1.5}Co_{0.5}Al_xFe_{16-x}O_{27}$ ($0.00 \leq x \leq 1.25$) are exhibited in Figure 3. Fourier transform infrared spectroscopy is a useful method for representing and recognizing chemical bonds, detecting functional groups, and detecting phase transitions at the tetrahedral and octahedral positions. The absorption bands in the frequency range of 400–800 $cm^{-1}$ correspond to stretching vibration at tetrahedral and octahedral positions in ferrite structures [23]. The absorption band at about 451 $cm^{-1}$ is attributed to the stretching vibration of octahedral metal ions and oxygen bonds, and the absorption band at about 594 $cm^{-1}$ is attributed to the stretching vibration of tetrahedral metal ions and oxygen bonds [24]. The absorption bands at the tetrahedral and octahedral sites showed an intrinsic deviation in behavior, and this may be due to the $Al^{3+}$ (0.535 Å) ion substitution for the $Fe^{3+}$ (0.645 Å) ions. It is obvious from the diagram that the normal vibrational modes of the tetrahedral clusters were higher than those of the octahedral clusters. This is mainly because the bond length of tetrahedral clusters is shorter than that of octahedral clusters [25]. The absorption band at about 673 $cm^{-1}$ in the spectrum of W-type SrYb hexaferrites with the Al content (x) = 0 could be due to the metal–oxygen stretching vibrations of $\alpha$-$Fe_2O_3$ [26]. The absorption band at about 1500 $cm^{-1}$ in the spectrum of W-type SrYb hexaferrites with the Al contents (x) = 0.5 and 1.0 may be caused by some trace impurities that entered the samples during the mixing of W-type SrYb hexaferrite powder with potassium bromide powder and the subsequent pressing of a very thin pancake sample. The spectra of W-type SrYb hexaferrites with different Al contents (x) indicate that the characteristic absorption bands at about 2921 $cm^{-1}$ and about 1628 $cm^{-1}$ are attributed to the presence of the O-H stretching bond because of the moisture content of W-type SrYb hexaferrites during their preparation process [27].

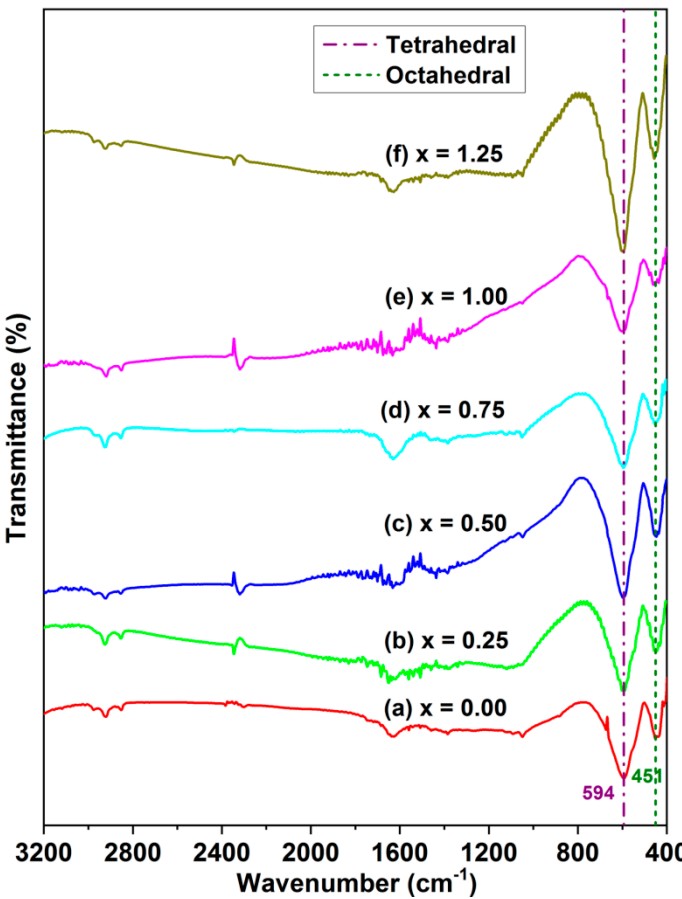

**Figure 3.** FT-IR spectra for the W-type SrYb hexaferrite $Sr_{0.85}Yb_{0.15}Zn_{1.5}Co_{0.5}Al_xFe_{16-x}O_{27}$ ($0.00 \leq x \leq 1.25$).

### 3.3. FE-SEM Images

The FE-SEM images of the W-type SrYb hexaferrite $Sr_{0.85}Yb_{0.15}Zn_{1.5}Co_{0.5}Al_xFe_{16-x}O_{27}$ ($0.00 \leq x \leq 1.25$) are shown in Figure 4. Figure 4a,b exhibit the FE-SEM fracture surface of the W-type SrYb hexaferrite pellets with the Al content (x) of x = 0.00 and x = 0.25, respectively. The lamellar structure of W-type SrYb hexaferrites can be clearly seen from the lamellar traces left by the fracture surface of the FE-SEM images of these W-type SrYb hexaferrites. Figure 4c–f present the FE-SEM images of the W-type SrYb hexaferrites with the Al contents (x) of x = 0.50, x = 0.75, x = 1.00, and x = 1.25, respectively. It is evident from the FE-SEM images that the morphology of the grains had no obvious change with the change in the Al content (x), and the prepared W-type SrYb hexagonal ferrites exhibited a transparent hexagonal plate.

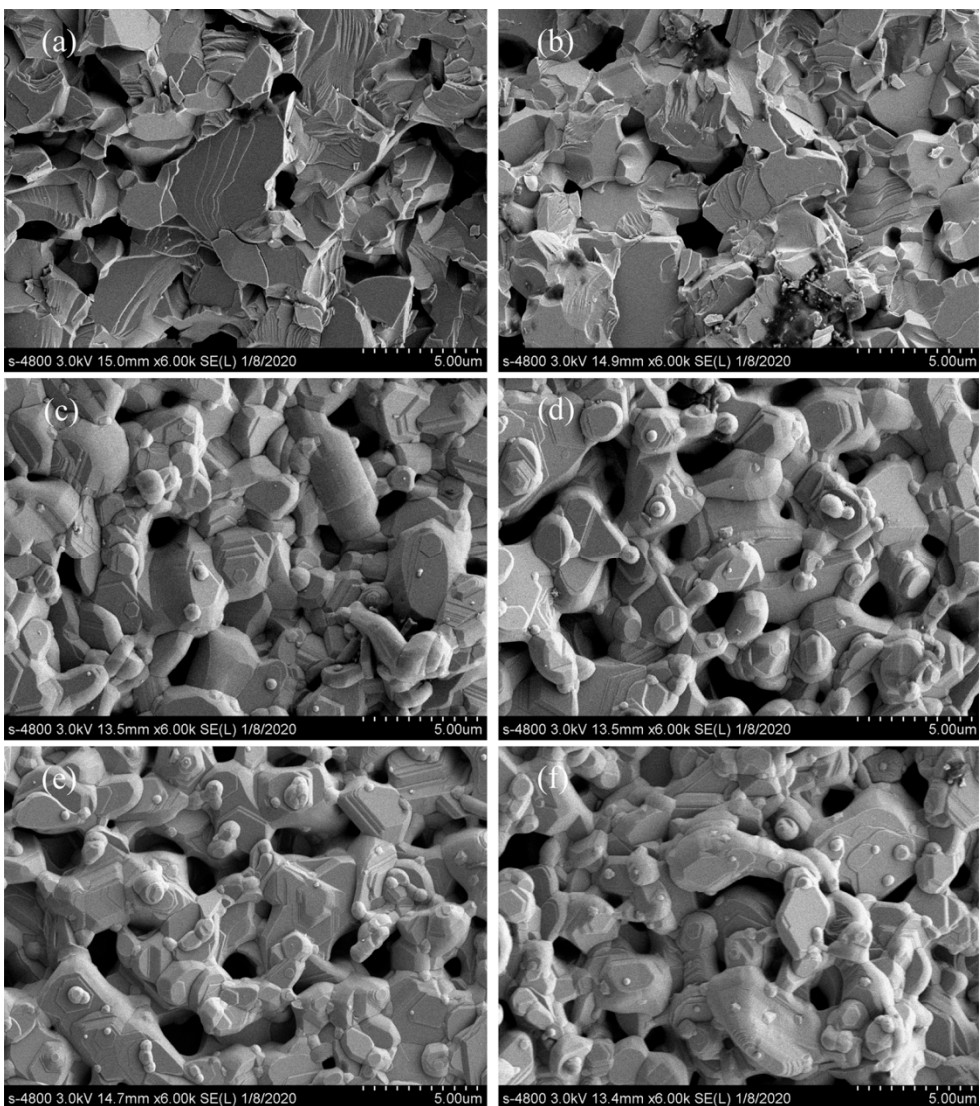

**Figure 4.** FE-SEM fracture surface of the W-type SrYb hexaferrite $Sr_{0.85}Yb_{0.15}Zn_{1.5}Co_{0.5}Al_xFe_{16-x}O_{27}$ with Al contents (x) of (**a**) x = 0.00 and (**b**) x = 0.25, and FE-SEM images of the W-type SrYb hexaferrite $Sr_{0.85}Yb_{0.15}Zn_{1.5}Co_{0.5}Al_xFe_{16-x}O_{27}$ with Al contents (x) of (**c**) x = 0.50, (**d**) x = 0.75, (**e**) x = 1.00, and (**f**) x = 1.25.

### 3.4. EDX Analysis

Figures 5 and 6 illustrate the EDX spectra and elemental mapping images of the W-type SrYb hexaferrite $Sr_{0.85}Yb_{0.15}Zn_{1.5}Co_{0.5}Al_xFe_{16-x}O_{27}$ with the Al content (x) ranging from x = 0.00 to x = 0.50, and the Al content (x) ranging from x = 0.75 to x = 1.25, respectively. As shown in Figures 5 and 6, the suitability of replacing Al for Fe in W-type SrYb hexaferrites is confirmed. The EDX spectra in Figures 5 and 6 for W-type SrYb hexaferrites with different Al contents (x) reveal the presence of Sr, Yb, Zn, Co, Al, Fe, and O, and their contents, which are in close agreement with the nominal elemental compositions. The inset of each figure exhibits the molecular and atomic weight of the element that was used in the preparation process. The increase in Al content and decrease in the host elements mean that the materials keep their stoichiometric contents. This confirms that Al ions replace Fe ions in the W-type SrYb hexaferrites.

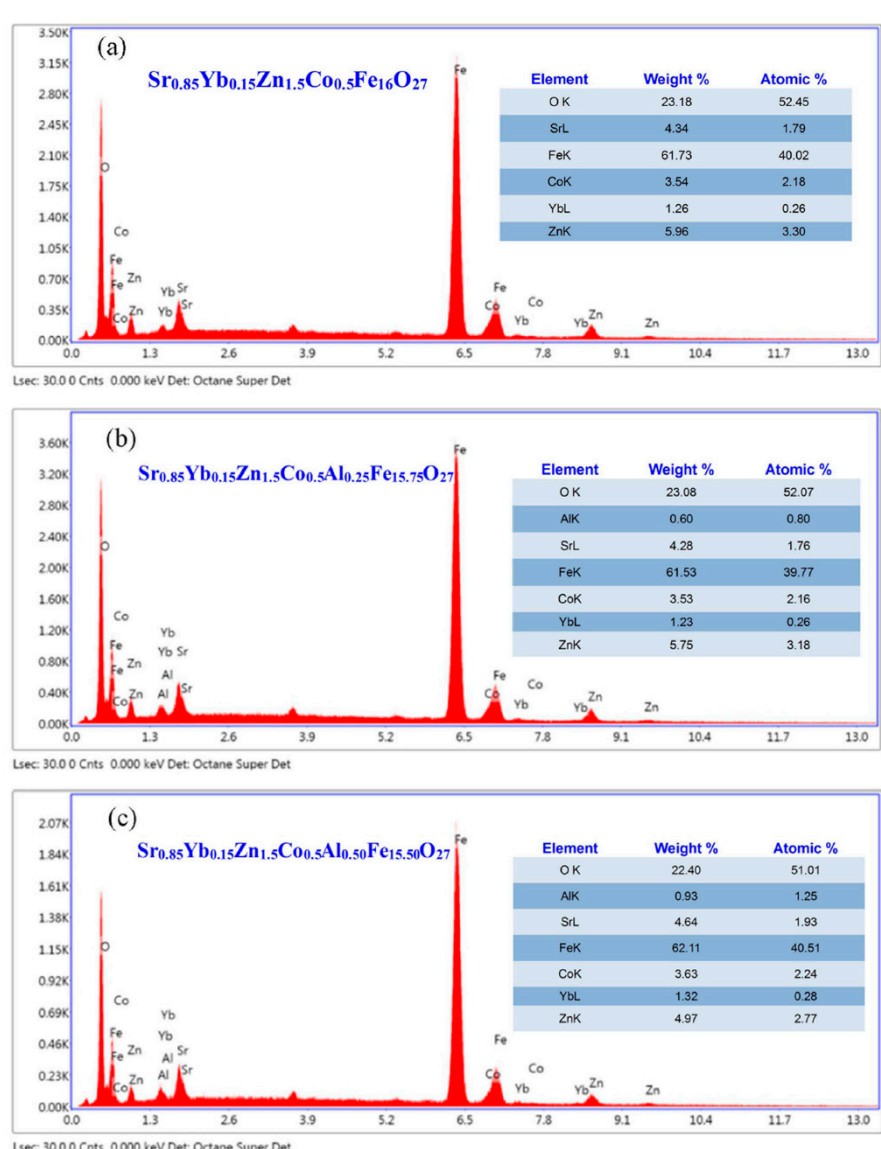

**Figure 5.** EDX spectra and elemental mapping images of the W-type SrYb hexaferrite $Sr_{0.85}Yb_{0.15}Zn_{1.5}Co_{0.5}Al_xFe_{16-x}O_{27}$ with Al content (x) of (**a**) x = 0.00, (**b**) x = 0.25, and (**c**) x = 0.50.

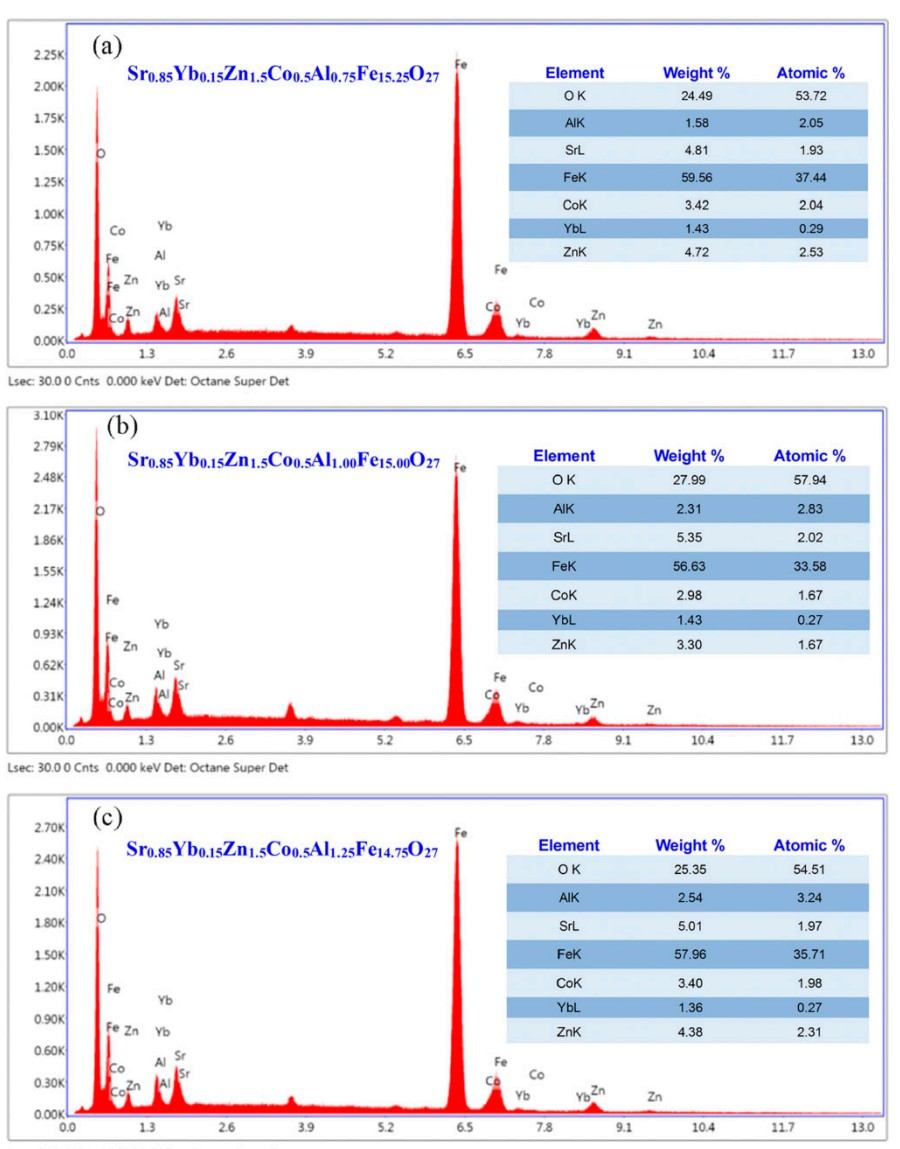

**Figure 6.** EDX spectra and elemental mapping images of the W-type SrYb hexaferrite $Sr_{0.85}Yb_{0.15}Zn_{1.5}Co_{0.5}Al_xFe_{16-x}O_{27}$ with Al content (x) of (**a**) x = 0.75, (**b**) x = 1.00, and (**c**) x = 1.25.

### 3.5. Magnetic Properties

The magnetic hysteresis loops with the maximum magnetic field of $\pm24000$ Oe, and the magnified charts of magnetic hysteresis loops within the range of $\pm750$ Oe, for the W-type SrYb hexaferrite $Sr_{0.85}Yb_{0.15}Zn_{1.5}Co_{0.5}Al_xFe_{16-x}O_{27}$ are exhibited in Figure 7a,b, respectively. From the magnetic hysteresis loops, the saturation magnetization ($M_s$), remanent magnetization ($M_r$), and coercivity ($H_c$) for W-type SrYb hexaferrites with the Al content (x) ranging from 0.00 to 1.25 were calculated.

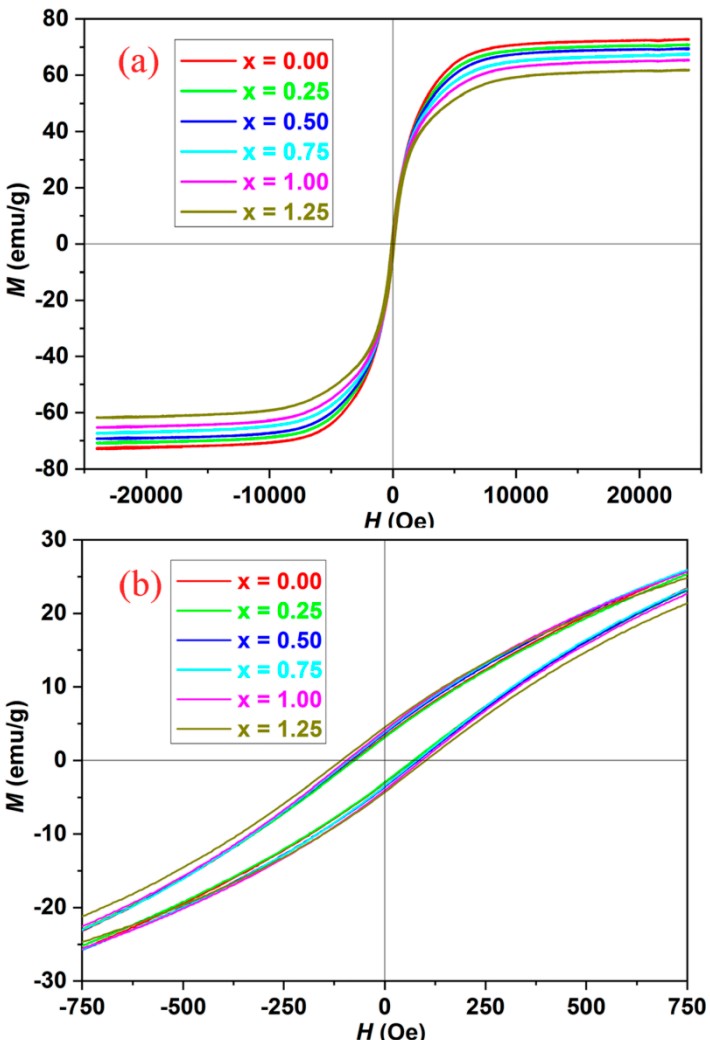

**Figure 7.** (**a**) Magnetic hysteresis loops with the maximum magnetic field of $\pm24{,}000$ Oe, and (**b**) the magnified charts of magnetic hysteresis loops within the range of $\pm750$ Oe, for the W-type SrYb hexagonal ferrite $Sr_{0.85}Yb_{0.15}Zn_{1.5}Co_{0.5}Al_xFe_{16-x}O_{27}$.

Figure 8 illustrates the effects of the Al content (x) on the saturation magnetization ($M_s$) and remanent magnetization ($M_r$) for the W-type SrYb hexaferrite $Sr_{0.85}Yb_{0.15}Zn_{1.5}Co_{0.5}Al_xFe_{16-x}O_{27}$. As can be seen from Figure 8, the value of $M_s$ gradually decreased from 72.779 emu/g at x = 0.00 to 61.882 emu/g at x = 1.25. Additionally, $M_r$ slightly decreased from 3.161 emu/g at x = 0.00 to 3.058 emu/g at x = 0.25 and increased by degrees as the Al content (x) increased from 0.25 to 1.25. This changing trend of $M_s$ is basically consistent with that of $BaCo_2Fe_{16-x}Al_xO_{27}$ synthesized using the traditional sol–gel method, which was studied by Ahmad et al. [16]. In the unit cell of a W-type hexaferrite that is composed of two hexagonal blocks (R block) and four spinel blocks (S block), there are seven kinds of crystal sites, detailed as follows: one trigonal bipyramidal site (2d), two tetrahedral sites (4 e and 4 $f_{IV}$), and four octahedral sites (12 k, 4 $f_{VI}$, 4 f, and 6 g) [28]. In W-type hexaferrites, the spin directions of the 12 k, 4 f, 6 g, and 2 d sublattices are spin-up, whereas the spin directions of the 4 e, 4 $f_{VI}$, and 4 $f_{IV}$ sublattices are spin-down, with the nearest cations coupled via superexchange interactions through the $O^{2-}$ ions [28]. The total magnetic moment (i.e., 27.4 $\mu_B$) of W-type hexaferrites is due to their uncompensated upward spins [16]. Thus, the magnetic properties of W-type hexaferrites can be affected by the occupation of their sites by different cations [12,14,16]. The magnetic moments of $Fe^{3+}$ and $Al^{3+}$ ions are 5.0 $\mu_B$ and about 0.0 $\mu_B$, respectively [16,29]. As the Al content (x) increased from 0.00 to 1.25, $M_s$ decreased, which is attributed to the following factor: Albanese et al. studied the

Al-substituted $Zn_2$-W hexaferrites and Al-substituted M-type Ba hexaferrites and found that $Al^{3+}$ ions preferentially enter into the octahedral sites 2 a and 12 k [30,31]. The non-magnetic $Al^{3+}$ ions having a magnetic moment of 0.0 $\mu_B$, which is less than that of $Fe^{3+}$ ion (5.0 $\mu_B$), replace the iron ions from the sites having spins in an upward direction, and this results in a decrease in the total magnetic moment. Therefore, the values of the saturation magnetization ($M_s$) decrease in the Al-substituted W-type SrYb hexaferrites.

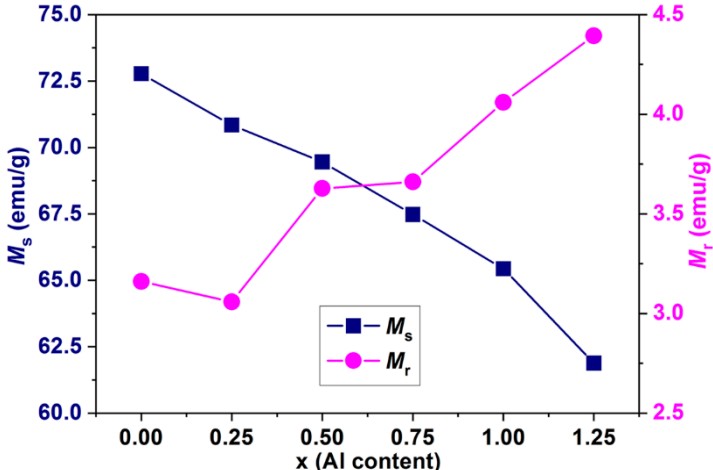

**Figure 8.** Effects of Al content (x) on the saturation magnetization ($M_s$) and remanent magnetization ($M_r$) of the W-type SrYb hexaferrite $Sr_{0.85}Yb_{0.15}Zn_{1.5}Co_{0.5}Al_xFe_{16-x}O_{27}$.

The Bohr magneton number ($n_B$) was calculated for the W-type SrYb hexaferrite $Sr_{0.85}Yb_{0.15}Zn_{1.5}Co_{0.5}Al_xFe_{16-x}O_{27}$ ($0.00 \leq x \leq 1.25$) using the relation given below [32]:

$$n_B = \frac{M_W \times M_S}{5585} \qquad (2)$$

where $M_W$ stands for the molecular weight. The values of the $M_r/M_s$ ratio for the W-type SrYb hexaferrites are calculated from $M_r$ and $M_s$. The effects of the Al content (x) on the magneton number ($n_B$) and $M_r/M_s$ of the W-type SrYb hexaferrite $Sr_{0.85}Yb_{0.15}Zn_{1.5}Co_{0.5}Al_xFe_{16-x}O_{27}$ are presented in Figure 9. From Figure 9, it is found that as the Al content (x) increased, the value of $n_B$ decreased, from 20.245 $\mu_B$ at x = 0.00 to 16.814 $\mu_B$ at x = 1.25. The trend of the Bohr magneton number ($n_B$) is in agreement with that of $M_s$, as shown in Figure 8. This means that the magnetic moment is responsible for the change in $M_s$. Additionally, the $M_r/M_s$ ratio is called the squareness ratio. As seen from Figure 9, the value of the $M_r/M_s$ ratio basically remained constant at about 0.0434 as the Al content (x) increased from 0.00 to 0.25 and then gradually increased from 0.0434 at x = 0.25 to 0.0710 at x = 1.25. These results indicate that the W-type SrYb hexaferrites with the Al content (x) ($0.00 \leqq x \leqq 1.25$) have a multidomain structure.

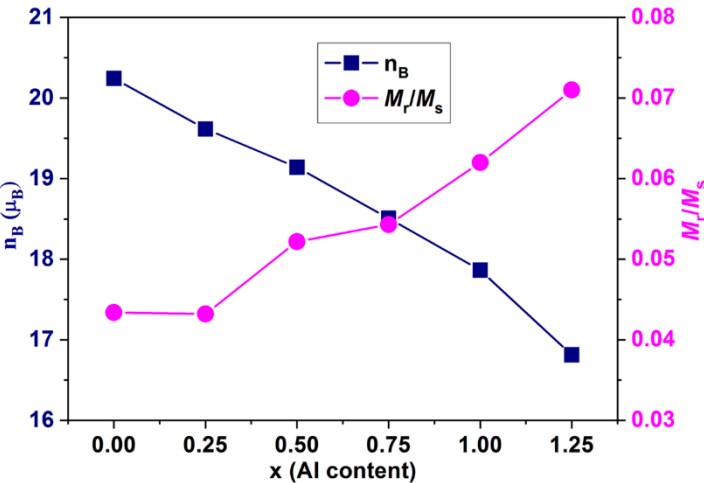

**Figure 9.** Effects of Al content (x) on the magneton number ($n_B$) and $M_r/M_s$ for the W-type SrYb hexaferrite $Sr_{0.85}Yb_{0.15}Zn_{1.5}Co_{0.5}Al_xFe_{16-x}O_{27}$.

The magnetic anisotropy field ($H_a$) and magnetocrystalline anisotropy constant ($K_1$) were determined through the law of saturation approach [33]. The relationship between the magnetization ($M$) and high magnetic fields ($H$) can be expressed in the following formula [34]:

$$M = M_S\left(1 - \frac{A}{H} - \frac{B}{H^2}\right) + \chi H \tag{3}$$

Here, $A$ is the constant caused by inhomogeneity, H is the external magnetic field, and X is susceptibility. In a strong magnetic field, constant $A$ is approximately zero, and $\chi$ is ignored. Therefore, Equation (3) is simplified to the following equation [35]:

$$M = M_S\left(1 - \frac{B}{H^2}\right) \tag{4}$$

For a strong magnetic field, the plots of $M$ vs. $1/H^2$ should give a straight line, as evident from Equation (4). Figure 10 exhibits the plots of the magnetization ($M$) versus $1/H^2$ of the W-type SrYb hexaferrite $Sr_{0.85}Yb_{0.15}Zn_{1.5}Co_{0.5}Al_xFe_{16-x}O_{27}$ with the Al content (x) ($0.00 \leqq x \leqq 1.25$). The slope gives the value of B, which denotes the magnetocrystalline anisotropy constant. Additionally, $K_1$ represents the first anisotropy constant, which can be calculated using Equation (5), and the magnetic anisotropy field ($H_a$) is calculated from the value of $K_1$ using Equation (6) [36]:

$$K_1 = M_S\left(\frac{15B}{4}\right)^{0.5} \tag{5}$$

$$H_a = \frac{2K_1}{M_S} \tag{6}$$

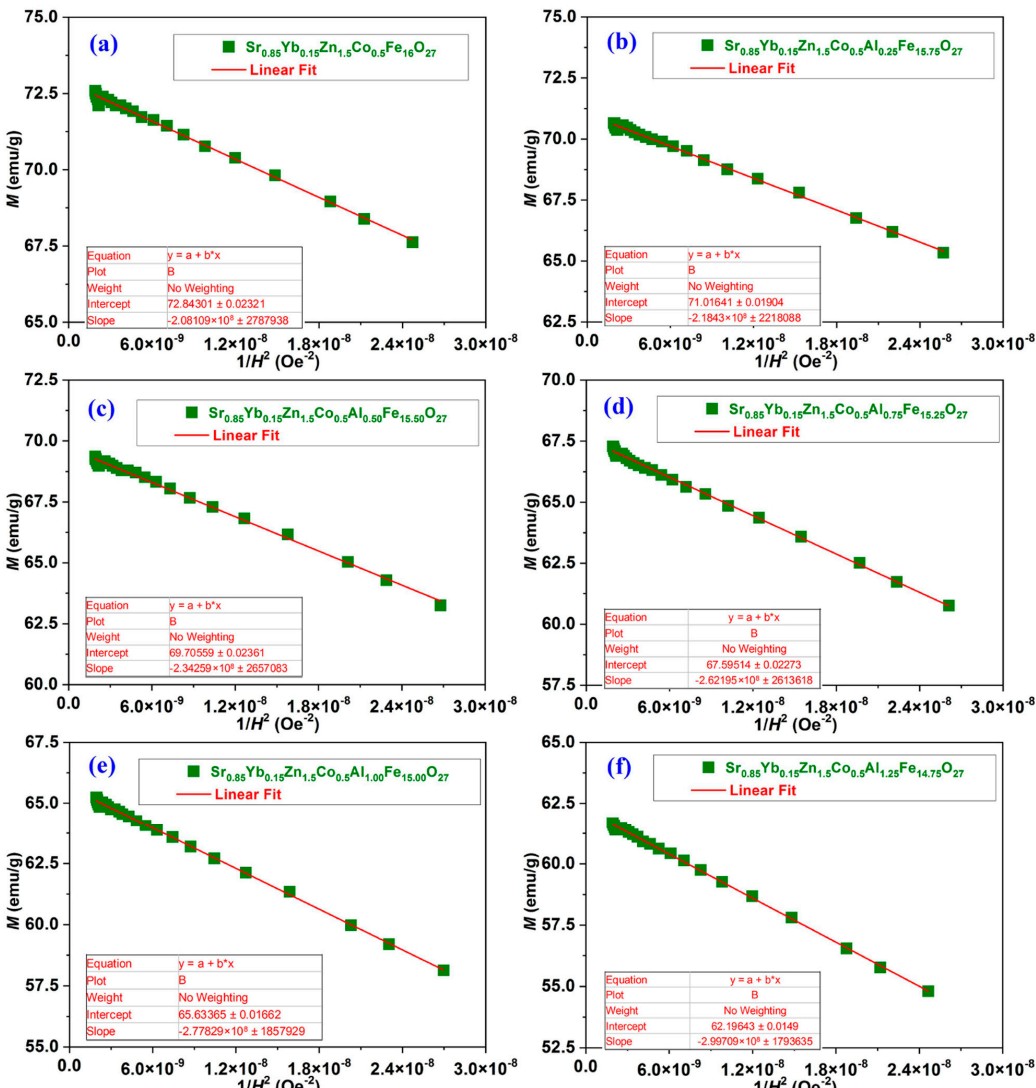

**Figure 10.** Plots of magnetization ($M$) versus $1/H^2$ for the W-type SrYb hexaferrite $Sr_{0.85}Yb_{0.15}Zn_{1.5}Co_{0.5}Al_xFe_{16-x}O_{27}$ with Al content (x) of (**a**) x = 0.00, (**b**) x = 0.25, (**c**) x = 0.50, (**d**) x = 0.75, (**e**) x = 1.00, and (**f**) x = 1.25.

The effects of the Al content (x) on the first anisotropy constant ($K_1$) and magnetic anisotropy field ($H_a$) of the W-type SrYb hexaferrite $Sr_{0.85}Yb_{0.15}Zn_{1.5}Co_{0.5}Al_xFe_{16-x}O_{27}$ ($0.00 \leq x \leq 1.25$) are presented in Figure 11. As seen from this figure, the value of $K_1$ increased from $2.384 \times 10^5$ erg/cm$^3$ at x = 0.00 to $2.644 \times 10^5$ erg/cm$^3$ at x = 1.25. Additionally, the value of $H_a$ increased from 6.546 kOe at x = 0.00 to 8.502 kOe at x = 1.25. Compared with Figure 9, the changing trends of the first anisotropy constant ($K_1$) and magnetic anisotropy field ($H_a$) with the increase in the Al content (x) from x = 0.00 to x = 1.25 are basically consistent with those of the $M_r/M_s$ ratio, which is called the squareness ratio. The increase in the value of $H_a$ can be attributed to the following fact: From Figure 8, it is evident that the value of $M_s$ gradually decreased with an increase in the Al content (x) from x = 0.00 to x = 1.25, while based on Figure 11, it is deduced that the value of $K_1$ increased with an increase in the Al content (x) from x = 0.00 to x = 1.25. According to Equation (6), $H_a$ is inversely proportional to $M_s$ and positively proportional to $K_1$. Therefore, the increase in the Al content (x) from x = 0.00 to x = 1.25 resulted in the increase in $H_a$.

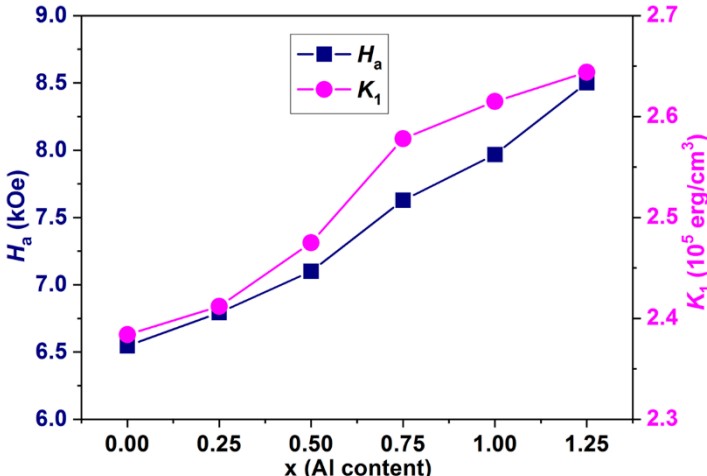

**Figure 11.** The impacts of Al content ($x$) on the first anisotropy constant ($K_1$) and magnetic anisotropy field ($H_a$) of the W-type SrYb hexaferrite $Sr_{0.85}Yb_{0.15}Zn_{1.5}Co_{0.5}Al_xFe_{16-x}O_{27}$ ($0.00 \leq x \leq 1.25$).

Figure 12 depicts the influence of the Al content ($x$) on the coercivity ($H_c$) of the W-type SrYb hexaferrite $Sr_{0.85}Yb_{0.15}Zn_{1.5}Co_{0.5}Al_xFe_{16-x}O_{27}$. As seen from Figure 12, with the increase in the Al content ($x$), the value of $H_c$ slightly decreased from 75 Oe at $x = 0.00$ to 73 Oe at $x = 0.25$ and then gradually increased from 73 Oe at $x = 0.25$ to 105 Oe at $x = 1.25$. All the samples had Hc values ranging from 70 Oe to 110 Oe. This low coercivity makes them ideal for microwave equipment, safety, switching, and sensing applications [37]. The trend in $H_c$ is not in agreement with that of $BaCo_2Fe_{16-x}Al_xO_{27}$, which was studied by Ahmad et al. [16]. However, the $H_c$ values in this study are slightly lower than those reported by Ahmad et al. [16]. The main reason is that the preparation method used in this study is different from that of the previous study [16]. The coercivity mechanism hinges on the internal and external properties of the material. The intrinsic properties of the material are related to its crystal structure and chemical composition, such as $M_s$, $T_c$, anisotropic energy $E_k$, and anisotropic field $H_a$. The external properties of the material are related to its microstructures, such as defects, grain size, and grain shape. In theory, the coercivity ($H_c$) of the particle assembly can be expressed in the following formula [38]:

$$H_c = \alpha \left( \frac{2K_1}{M_s} - N_d M_S \right) = \alpha(H_a - H_d) \tag{7}$$

where $\alpha$ is a constant, $N_d$ is the shape-demagnetizing factor, and $H_d$ is the shape-demagnetizing anisotropy field. As shown in Figure 4, the platelet-like shape and the average grain size basically remained unchanged with the Al substitution, and thus, $N_d$ remained basically the same. From Equation (7), the increase in coercivity ($H_c$) with the Al content ($x$) increasing from 0.25 to 1.25 mainly arises from the increase in the magnetic anisotropy field ($H_a$), as shown in Figure 11. According to the coercivity mechanism, the slight decrease in the value of $H_c$, from 75 Oe at $x = 0.00$ to 73 Oe at $x = 0.25$, is mainly due to the external properties of the material, such as slight changes in defects, grain size, and grain shape [16].

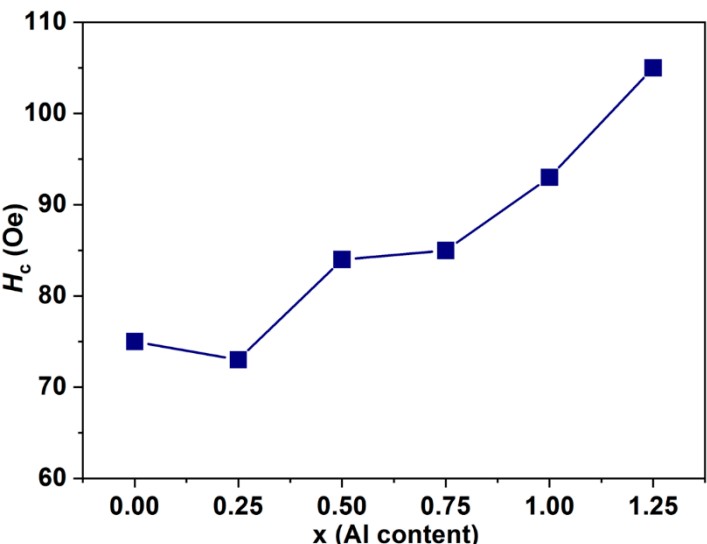

**Figure 12.** The impact of Al content (x) on the coercivity ($H_c$) of the W-type SrYb hexaferrite $Sr_{0.85}Yb_{0.15}Zn_{1.5}Co_{0.5}Al_xFe_{16-x}O_{27}$.

## 4. Conclusions

In this study, the modified ceramic method, i.e., the two-step calcination method, was used to synthesize Al-substituted W-type SrYb hexaferrites with the nominal composition $Sr_{0.85}Yb_{0.15}Zn_{1.5}Co_{0.5}Al_xFe_{16-x}O_{27}$ ($0.00 \leq x \leq 1.25$). The microstructures, spectral bands of characteristic functional groups, morphologies, and magnetic parameters of the prepared samples were characterized using XRD, FTIR, SEM, EDX, and VSM. The XRD results show that, compared with the standard patterns for the W-type hexaferrite, the W-type SrYb hexaferrites with the Al content (x) of $0.00 \leq x \leq 1.25$ are a single W-type hexaferrite phase. The absorption band at about 451 cm$^{-1}$ is attributed to the stretching vibration of the octahedral metal ions and oxygen bonds, and the absorption band at about 594 cm$^{-1}$ is attributed to the stretching vibration of the tetrahedral metal ions and oxygen bonds. As shown in SEM images, the grains of the W-type hexaferrites with various Al contents (x) have hexagonal platelet-like shapes. The EDX spectra for the W-type SrYb hexaferrites with the Al content (x) ($0.00 \leqq x \leqq 1.25$) revealed the presence and contents of Sr, Yb, Zn, Co, Al, Fe, and O and are closely aligned with the nominal elemental compositions. $M_s$ and $n_B$ decreased with the Al content (x) increasing from 0.00 to 1.25. $M_r$ and $H_c$ decreased with the Al content (x) increasing from 0.00 to 0.25. Additionally, when the Al content (x) $\geq 0.25$, $M_r$ and $H_c$ increased with an increase in the Al content (x). $H_a$ and $K_1$ increased with the Al content (x) increasing from 0.00 to 1.25. The Al-substituted W-type SrYb hexaferrites exhibited a soft magnetic behavior because of their high saturation magnetization and low coercivity; thus, they may be used as microwave-absorbing materials.

**Funding:** This research received no external funding.

**Institutional Review Board Statement:** Not applicable.

**Informed Consent Statement:** Not applicable.

**Data Availability Statement:** Not applicable.

**Conflicts of Interest:** The author declares that he has no known competing financial interests or personal relationships that could have appeared to influence the work reported in this paper.

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
