# Peer review of "Two-Step Calcination-Method-Derived Al-Substituted W-Type SrYb Hexaferrites: Their Microstructural, Spectral, and Magnetic Properties"

_magnetochemistry, doi:10.3390/magnetochemistry8100118_

Round 1
Reviewer 1 Report
This work present two-step calcination method derived Al substituted W-type SrYb hexaferrites: the microstructural, spectral and magnetic properties. The authors discuss the method of synthesis of the materials, their phase purity and morphology. They also investigate the magnetic properties of the material influenced by Al substitution for Fe.
However, some critical issues should be raised:
1) The authors should show XRD investigation after each step of annealing and to make clear on which samples is made it- pellets or powders.
2) The authors must give more information about the " fracture surface " of the samples. What they look like- powder, bulk materials or some mix state?
3) The authors must give more detailed explanation for the variation of the coercivity with changing of the Al content.
Author Response
Response to Reviewer 1 Comments
Comments and Suggestions for Authors
This work present two-step calcination method derived Al substituted W-type SrYb hexaferrites: the microstructural, spectral and magnetic properties. The authors discuss the method of synthesis of the materials, their phase purity and morphology. They also investigate the magnetic properties of the material influenced by Al substitution for Fe.
However, some critical issues should be raised:
- The authors should show XRD investigation after each step of annealing and to make clear on which samples is made it- pellets or powders.
Response 1): To this comment, thank you very much for your suggestion. When I done the experiemnts, all samples were two-step calcined, and the samples that were calcined on the first step was not reserved. And thus the XRD investigation on the first step of annealing can not be redone. In this paper, all XRD investigation was done on the powders that are obtained by crushing the two-step calcined pellets by vibration mill. And I have added the corresponding explanation in the article as can be seen in the manuscript.
- The authors must give more information about the " fracture surface " of the samples. What they look like- powder, bulk materials or some mix state?
Response 2): To this comment, thank you very much for your suggestion. have added more information about the "fracture surface" of the samples in the article as can be seen in the manuscript.
3) The authors must give more detailed explanation for the variation of the coercivity with changing of the Al content.
Response 3): To this comment, thank you very much for your suggestion. I have added more detailed explanation for the variation of the coercivity with changing of the Al content in the article as can be seen in the manuscript.

Reviewer 2 Report
In the present manuscript the authors have synthesized for the first time the Al substituted W-type SrYb hexaferrites with nominal composition 286 Sr0.85Yb0.15Zn1.5Co0.5AlxFe16-xO27 (0.00 ≤ x ≤ 1.25).using the modified ceramic method, i. e. two-step calcination method. They have investigated these materials using various experimental techniques, such as XRD, FTIR, SEM, EDX and VSM. The XRD and SEM/EDX studies show that Al-substituted samples are single phase. The lattice a and c decrease slightly with increasing x indicating that Al is going on Fe sites. Further, FTIR spectra show the characteristic absorption bands at about 2921 cm−1 and about 1628 cm^-1I s attributed to the presence of O-H stretching bond because of the moisture content of the W-type SrYb hexaferrites during the preparation process. The saturation magnetization (Ms) and magneton number (nB) decreased with Al content (x) from 0.00 to 1.25. The remanent magnetization (Mr) and coercivity (Hc) decreased with Al content (x) from 0.00 to 0.25. And when Al content (x) ≥ 0.25, Mr and Hc increased with increasing Al content (x). The magnetic anisotropy field (Ha) and first anisotropy constant (K1) increased with Al content (x) from 0.00 to 1.25.
The manuscript is on interesting material and contains new results. The experimental results are high quality and will be useful for the researcher working in the field of ferrites. I recommend the manuscript to be published after the authors have considering following minor comments.
Comments/suggestions
[1] In Fig.2 right y-axis (numbers and title) may be used with same colour as the symbols.
[2] In Fig.3 For x=0 red colour curve there is a spike (near 700 cm^-1) above 594 cm^-1 absorption, which is not there in other samples. Please clarify this in the revised manuscript.
[3] Also in Fig.3 for x=0.5 and 1.0, there is a clear peak near 1500 cm^-1. What is origin for this?
[4] It would be good to add more information in the conclusion section.
Once authors have included these corrections/suggestions the manuscript can be published in Magnetochemistry.
Author Response
Response to Reviewer 2 Comments
Comments and Suggestions for Authors
In the present manuscript the authors have synthesized for the first time the Al substituted W-type SrYb hexaferrites with nominal composition 286 Sr0.85Yb0.15Zn1.5Co0.5AlxFe16-xO27 (0.00 ≤ x ≤ 1.25).using the modified ceramic method, i. e. two-step calcination method. They have investigated these materials using various experimental techniques, such as XRD, FTIR, SEM, EDX and VSM. The XRD and SEM/EDX studies show that Al-substituted samples are single phase. The lattice a and c decrease slightly with increasing x indicating that Al is going on Fe sites. Further, FTIR spectra show the characteristic absorption bands at about 2921 cm−1 and about 1628 cm^-1I s attributed to the presence of O-H stretching bond because of the moisture content of the W-type SrYb hexaferrites during the preparation process. The saturation magnetization (Ms) and magneton number (nB) decreased with Al content (x) from 0.00 to 1.25. The remanent magnetization (Mr) and coercivity (Hc) decreased with Al content (x) from 0.00 to 0.25. And when Al content (x) ≥ 0.25, Mr and Hc increased with increasing Al content (x). The magnetic anisotropy field (Ha) and first anisotropy constant (K1) increased with Al content (x) from 0.00 to 1.25.
The manuscript is on interesting material and contains new results. The experimental results are high quality and will be useful for the researcher working in the field of ferrites. I recommend the manuscript to be published after the authors have considering following minor comments.
Comments/suggestions
[1] In Fig.2 right y-axis (numbers and title) may be used with same colour as the symbols.
Response 1): To this comment, thank you very much for your suggestion. In Fig.2 right y-axis (numbers and title) have been used with same colour as the symbols as can be seen in the manuscript.
[2] In Fig.3 For x=0 red colour curve there is a spike (near 700 cm-1) above 594 cm^-1 absorption, which is not there in other samples. Please clarify this in the revised manuscript.
Response 2): To this comment, thank you very much for your suggestion. I have tried to clarify the spike (near 700 cm-1) above 594 cm-1 absorption in the article as can be seen in the manuscript.
[3] Also in Fig.3 for x=0.5 and 1.0, there is a clear peak near 1500 cm-1. What is origin for this?
Response 3): To this comment, thank you very much for your suggestion. I have carefully checked the peaks of the hexaferrites of studies which many resrarchers have done, and I have not found any useful information connecting with the clear peak near 1500 cm-1. Therefore, I speculate that this peak may be caused by some trace impurities that I introduced in the preparation of the FTIR sample. That is, trace impurities enter the sample during the mixing of ferrite powder with potassium bromide powder and the subsequent pressing of a very thin pancake sample. I have added the presentation in the article as can be seen in the manuscript.
[4] It would be good to add more information in the conclusion section.
Response 4): To this comment, thank you very much for your suggestion. I have added more information in the conclusion section as can be seen in the manuscript.
Once authors have included these corrections/suggestions the manuscript can be published in Magnetochemistry.

Reviewer 3 Report
The manuscript is on topic in the field of synthesis and characterization of hexaferrites. This material is intensively studied for a new applications. The paper is well prepared in all sections. Synthesis and measurement methods are well described. Results are supported by experimental data. Conclusions are clearly presented. I have no further comments and suggestions.
Author Response
Response to Reviewer 3 Comments
Comments and Suggestions for Authors
The manuscript is on topic in the field of synthesis and characterization of hexaferrites. This material is intensively studied for a new applications. The paper is well prepared in all sections. Synthesis and measurement methods are well described. Results are supported by experimental data. Conclusions are clearly presented. I have no further comments and suggestions.
Response: Thank you very much for your affirmation and approval of my article.

Reviewer 4 Report
The article reports the synthesize of W-type SrYb hexagonal ferrites with different Al3+ substitution [Sr0.85Yb0.15Zn1.5Co0.5AlxFe16-xO27 (0.00 ≤ x ≤ 1.25)] by the two-step calcination method. The author characterized the prepared samples with XRD, FTIR, EDX, and SEM. The magnetic properties also were studied by VSM. I suggest that the manuscript can be considered for publication in Magnetochemistry after major revision.
The suggested improvement and corrections:
1- It will be better if the author adds promising applications for his samples in the abstract or conclusion section according to the measured properties in the article.
2- The abstract shows only results, Please add an introduction and summary.
3- In the introduction section, the author mentioned previous studies for W-type hexaferrite with different substitutions, so please add the main results of each of them in the introduction.
4- Please rewrite the final part in the introduction section without (We) because I saw only one author in the article.
5- Please delete this sentence from the materials and methods section “The fine powder with particle size ranging from 2μm to 5μm was obtained by crushing the calcined pellet by vibration mill ”. The author may rewrite the mentioned sentence without mentioning the particle size range (there is no need to mention this information in this section).
6- In line no. 76. From which company did the author purchase the chemicals?
7- The author doesn’t have any as-prepared ferrite to make the calcination process. (The word calcination doesn’t reflect the solid state reaction using oxides as a raw material). In line no. 78. Please replace the word calcination with sintering. I recommend changing the word calcination in the whole article and title into sintering.
8- It would be better if the author mentioned wt.% or mol.% after the sample composition studied in the article.
9- In line no. 89. Stress for crystal or the author means (to eliminate the crystal strain and lattice defects)
10- Please add references for the ionic radius of Al and Fe mentioned in the text (line no.122).
11- Can the author explain why the weight percentage of Fe is more than O in the EDX analysis?
12- Please explain why the anisotropy increase with increasing Al-substitution and the relation between the anisotropy and the squareness factor.
13- The novelty in the article is not clear. Please reconsider.

Author Response
Response to Reviewer 4 Comments
Comments and Suggestions for Authors
The article reports the synthesize of W-type SrYb hexagonal ferrites with different Al3+ substitution [Sr0.85Yb0.15Zn1.5Co0.5AlxFe16-xO27 (0.00 ≤ x ≤ 1.25)] by the two-step calcination method. The author characterized the prepared samples with XRD, FTIR, EDX, and SEM. The magnetic properties also were studied by VSM. I suggest that the manuscript can be considered for publication in Magnetochemistry after major revision.
The suggested improvement and corrections:
- It will be better if the author adds promising applications for his samples in the abstract or conclusion section according to the measured properties in the article.
Response 1): To this comment, thank you very much for your suggestion. I have added promising applications for the samples in theconclusion section according to the measured properties in the article as can be seen in the manuscript.
- The abstract shows only results, Please add an introduction and summary.
Response 2): To this comment, thank you very much for your suggestion. In the abstract, I have added the introduction and summary in the article as can be seen in the manuscript.
3- In the introduction section, the author mentioned previous studies for W-type hexaferrite with different substitutions, so please add the main results of each of them in the introduction.
Response 3): To this comment, thank you very much for your suggestion. I have added the main results of each of previous studies for W-type hexaferrite with different substitutions in the introduction of the article as can be seen in the manuscript.
4- Please rewrite the final part in the introduction section without (We) because I saw only one author in the article.
Response 4): To this comment, thank you very much for your suggestion. In the final part in the introduction section, I have changed the inappropriate expression as can be seen in the manuscript.
5- Please delete this sentence from the materials and methods section “The fine powder with particle size ranging from 2μm to 5μm was obtained by crushing the calcined pellet by vibration mill ”. The author may rewrite the mentioned sentence without mentioning the particle size range (there is no need to mention this information in this section).
Response 5): To this comment, thank you very much for your suggestion. I rewritten the mentioned sentence in the article as can be seen in the manuscript.
6- In line no. 76. From which company did the author purchase the chemicals?
Response 6): To this comment, thank you very much for your suggestion. I have added the company from which the author purchase the chemicals in the article as can be seen in the manuscript.
7- The author doesn’t have any as-prepared ferrite to make the calcination process. (The word calcination doesn’t reflect the solid state reaction using oxides as a raw material). In line no. 78. Please replace the word calcination with sintering. I recommend changing the word calcination in the whole article and title into sintering.
Response 7): To this comment, thank you very much for your suggestion. In the preparation of ferrite magnet by solid-state method, there are two processes, calcination (i.e.: pre-sintering) and sintering, which are as follows:
Firstly, all raw materials such as BaCO3, SrCO3, and Fe2O3 etc. were weighted in stoichimetric ratio. And then, the raw materials were thoroughly mixed together in water in a ball mill. Further, the as-mixed powder was dried, and pressed into circular pellets with Φ30 × 16 mm. Subsequently, the pellets were calcined in a muffle furnace at ≧1200 oC for about 2.0 h.
Secondly, the calcined pellets were shattered, and suitable additives (CaCO3 0.3wt%-1.0wt%, SiO2 0.1wt%-0.5wt% and Al2O3 0.1wt%-0.5wt% etc.) were added, then the mixture was wet-milled in a ball-mill. This procedure guarantees a narrow particle size distribution with the mean size of around 0.75μm. The fine milled hexaferrite slurry was pressed into circular pellets in a magnetic field of ≧0.8 T. Finally, all green pellets were sintered at about 1185 oC for about 1.5 h.
In this paper, I mainly study the calcination process, using calcination process divided into two completed. This article does not study the sintering process in the solid-state reaction, then I will do a detailed study of the sintering process. Therefore, the use of the word “calcination” in this article is more suitable for the research purposes of this article.
8- It would be better if the author mentioned wt.% or mol.% after the sample composition studied in the article.
Response 8): To this comment, thank you very much for your suggestion. I have added the important information, such as in the part of “2. Materials and Methods”: “In this work, SrCO3 (99.5wt.%) powder, Yb2O3 (99.9wt.%) powder, ZnO (99wt.%) powder, CoO (99wt.%) powder, Al2O3 (99wt.%) powder, Fe2O3 (99.3wt.%) powder were used as raw materials. ”; and “In this article, in nominal composition Sr0.85Yb0.15Zn1.5Co0.5AlxFe16-xO27 (0.00 ≤ x ≤ 1.25), the subscript of each metal element and oxygen is the ratio of the number of moles.” in the article as can be seen in the manuscript.
9- In line no. 89. Stress for crystal or the author means (to eliminate the crystal strain and lattice defects)
Response 9): To this comment, thank you very much for your suggestion. In line no. 89., I have Changed to your statement “to eliminate the crystal strain and lattice defects”in the article as can be seen in the manuscript.
10- Please add references for the ionic radius of Al and Fe mentioned in the text (line no.122).
Response 10): To this comment, thank you very much for your suggestion. I have added references for the ionic radius of Al and Fe mentioned in the text (line no.122) in the article as can be seen in the manuscript.
11- Can the author explain why the weight percentage of Fe is more than O in the EDX analysis?
Response 11): To this comment, thank you very much for your suggestion. The EDX spectra in Figure 5 and Figure 6 for the W-type SrYb hexaferrites with different Al content (x) enlightening the presence of Sr, Yb, Zn, Co, Al, Fe and O and their contents, are in close agreement with the nominal elemental compositions. The molecular numbers of iron and oxygen are 55.85 and 16, respectively. In this article, in nominal composition Sr0.85Yb0.15Zn1.5Co0.5AlxFe16-xO27 (0.00 ≤ x ≤ 1.25), if x=1.25, the moles of iron and oxygen were 14.75 and 27, respectively. Thus in nominal composition Sr0.85Yb0.15Zn1.5Co0.5AlxFe16-xO27 (0.00 ≤ x ≤ 1.25), if x=1.25, the weight of iron and oxygen is 826 g and 432 g, respectively. This is the reason for the weight percentage of Fe is more than O in the EDX analysis.
12- Please explain why the anisotropy increase with increasing Al-substitution and the relation between the anisotropy and the squareness factor.
Response 12): To this comment, thank you very much for your suggestion. I have explained why the anisotropy increase with increasing Al-substitution in page 12 in the part “3.5. Magnetic Properties” in the article as can be seen in the manuscript.
And the relation between the anisotropy and the squareness factor “Compared with Figure 9, the changing trends of the first anisotropy constant (K1) and magnetic anisotropy field (Ha) with the increase of Al content (x) from x = 0.00 to x = 1.25 are basically are in consistent with that of Mr/Ms ratio which is called the squareness ratio. ” has been added in page 12 in the part “3.5. Magnetic Properties” in the article as can be seen in the manuscript.
13- The novelty in the article is not clear. Please reconsider.
Response 13): To this comment, thank you very much for your suggestion. I have recondidered the novelty in the article. The novelty in the article is listed in the below:
Up to present, no prior reports have been carried out to investigate W-type Sr hexaferrites with Al3+ substitution for Fe3+ sites synthesized by the two-step calcination method. And this should be the novelty of this article. I have successfully synthesized Al substituted W-type SrYb hexaferrites Sr0.85Yb0.15Zn1.5Co0.5AlxFe16-xO27 (0.00 ≤ x ≤ 1.25) by the two-step calcination method. At the same time, I have systematically investigated the influence of Al substitution on the microstructural, spectral and magnetic properties of W-type SrYb hexaferrites.

Round 2
Reviewer 4 Report
According to the reviewer's comments, the author has significantly improved the manuscript, and now the article can be accepted for publication.